# History of Mammography: Analysis of Breast Imaging Diagnostic Achievements over the Last Century

**DOI:** 10.3390/healthcare11111596

**Published:** 2023-05-30

**Authors:** Luca Nicosia, Giulia Gnocchi, Ilaria Gorini, Massimo Venturini, Federico Fontana, Filippo Pesapane, Ida Abiuso, Anna Carla Bozzini, Maria Pizzamiglio, Antuono Latronico, Francesca Abbate, Lorenza Meneghetti, Ottavia Battaglia, Giuseppe Pellegrino, Enrico Cassano

**Affiliations:** 1Breast Imaging Division, Radiology Department, IEO European Institute of Oncology IRCCS, 20141 Milan, Italy; filippo.pesapane@ieo.it (F.P.); anna.bozzini@ieo.it (A.C.B.); maria.pizzamiglio@ieo.it (M.P.); antuono.latronico@ieo.it (A.L.); francesca.abbate@ieo.it (F.A.); lorenza.meneghetti@ieo.it (L.M.); enrico.cassano@ieo.it (E.C.); 2Postgraduation School of Diagnostic and Interventional Radiology, University of Milan, Via Festa del Perdono 7, 20122 Milan, Italy; giulia.gnocchi@unimi.it (G.G.); ottavia.battaglia@unimi.it (O.B.); giuseppe.pellegrino@unimi.it (G.P.); 3Centre of Research in Osteoarchaeology and Paleopathology, Department of Biotechnology and Life Sciences, University of Insubria, Via J.H. Dunant, 3, 21100 Varese, Italy; ilaria.gorini@uninsubria.it; 4Diagnostic and Interventional Radiology Department, Circolo Hospital, ASST Sette Laghi, 21100 Varese, Italy; massimo.venturini@uninsubria.it (M.V.); federico.fontana@uninsubria.it (F.F.); 5School of Medicine and Surgery, Insubria University, 21100 Varese, Italy; 6Radiology Department, Università degli Studi di Torino, 10129 Turin, Italy; ida.abiuso@unito.it

**Keywords:** history, breast cancer, mammography, digital breast tomosynthesis, contrast-enhanced mammography, breast cancer screening

## Abstract

Breast cancer is the most common forms of cancer and a leading cause of mortality in women. Early and correct diagnosis is, therefore, essential to save lives. The development of diagnostic imaging applied to the breast has been impressive in recent years and the most used diagnostic test in the world is mammography, a low-dose X-ray technique used for imaging the breast. In the first half of the 20th century, the diagnosis was in practice only clinical, with consequent diagnostic delay and an unfavorable prognosis in the short term. The rise of organized mammography screening has led to a remarkable reduction in mortality through the early detection of breast malignancies. This historical review aims to offer a complete panorama of the development of mammography and breast imaging during the last century. Through this study, we want to understand the foundations of the pillar of radiology applied to the breast through to the most modern applications such as contrast-enhanced mammography (CEM), artificial intelligence, and radiomics. Understanding the history of the development of diagnostic imaging applied to the breast can help us understand how to better direct our efforts toward an increasingly personalized and effective diagnostic approach. The ultimate goal of imaging applied to the detection of breast malignancies should be to reduce mortality from this type of disease as much as possible. With this paper, we want to provide detailed documentation of the main steps in the evolution of breast imaging for the diagnosis of breast neoplasms; we also want to open up new scenarios where the possible current and future applications of imaging are aimed at being more precise and personalized.

## 1. Introduction

Breast cancer is the most diagnosed tumor, with over two million new cases reported in 2020, and it is the leading cause of death by cancer in women [1]. Therefore, early and effective diagnosis has always been the most critical challenge for physicians fighting this type of disease.

The primary diagnostic method used to diagnose breast neoplasia is mammography. Mammography is an X-ray technique used for imaging the breast. It can detect breast lesions before they become palpable. In the past 100 years, we have seen a dramatic development in the diagnosis of breast malignancies. In particular, the advent of organized mammography screening in asymptomatic women has dramatically reduced mortality from breast malignancy [2]. Women who undergo mammography screening examinations have been shown to have an approximately 30% reduction in mortality compared with women who do not undergo screening [3].

With this work, we will review the history of breast imaging, describing its development through to the most modern applications. Understanding this progress can help us to understand the pathology we face in the present and the importance of the correct diagnostic balance in the management of the patient with breast cancer.

The purpose of our review is to provide detailed documentation of the main stages of the development of breast imaging, to facilitate discussion of the latest methods that, in the coming years, can offer increasingly efficient and personalized management of breast neoplasms.

## 2. Origin of Mammography

Although X-rays were first discovered [4] in 1895 in Würzburg (Germany) by Rontgen (1845–1923), the first mammogram was only performed in 1913 by the German surgeon Albert Solomon (1883–1976). In his paper “*Contributions to the Pathology in Clinical Medicine of Breast Cancer”*, he demonstrated the existence of different types of breast cancer and their spreading through the axillary lymph nodes by performing an X-ray study on 3000 mastectomies [5]. Despite his discoveries, the outbreak of World War I put a stop to his efforts, and several years passed before the X-ray evaluation of the breast in a living person was finally introduced.

In 1927, German surgeon Otto Kleinschmidt (1880–1948) and his mentor Erwin Payr described the first mammography and its crucial role in the early detection of breast cancer [6]. A similar paper analyzing the role of radiology in the assessment of breast tumors was published in 1932 by Walter Vogel [7].

Shortly after, in 1930, Stafford Warren (1896–1981) performed the first mammogram in the United States. The radiologist analysed the potential role of the technique in the preoperative assessment of breast tumors. Warren described the regular appearance of the mammary gland and stressed the need to examine both breasts, as the comparison between the two allowed better identification of abnormalities [8].

Starting from the late 1930s, the radiologist Gershon-Cohen (1899–1971) published several studies describing the variability of the normal radiographic appearance of the breast [9,10]. His work took a big step forward when he started his collaboration with the pathologist Helen Ingleby (1890–1973), shedding new light on the understanding of breast malignancies, thanks to the correlation between the pathologic and roentgenographic aspects of breast diseases. In their “*Comparative Anatomy, Pathology, and Roentgenology of the Breast”*, they also analyzed the congenital abnormalities of the breast and its normal variation according to age, menstrual status, pregnancy, and lactation, as the knowledge of what is normal is a prerequisite to understanding what is pathological [11].

Another pioneer in the evolution of mammography was Raul Leborgne (1907–1986), a radiologist from Uruguay. In his papers in 1951, he was the first researcher who focused on the importance of microcalcifications to discriminate between benign and malignant diseases; their patterns and characteristics, as seen in mammography, allowed the identification of 30–50% of non-palpable breast cancers [12]. He developed the breast compression technique to improve image quality and lower the average radiation exposure of the breast [13]. His intuitions are now part of the standard clinical practice of mammography execution.

The public awareness of cancer took a big step forward after World War II; breast cancer was recognized as the second leading cause of death among women, emphasizing the need to obtain a better and earlier diagnosis. However, it was only in the late 1960s that mammography started to spread as a possible screening tool.

## 3. Technical Progress

Before the 1950s, many devices used to study the breast were not dedicated to the breast itself—X-ray units were intended for other diagnostic procedures such as chest X-rays. Therefore, the results were sub-optimal in terms of both breast compression and focal spots, resulting in significant dose exposure and inadequate image quality [14].

Dr. Roach and Dr. Hilliboe, in 1952, highlighted the medical potential of xeroradiography [15]. Some semiconductors, such as selenium, become charge conductors when stimulated by ionizing radiation—this is the process behind xeroradiography [16]. In 1960, Gould et al. [17] demonstrated that it was possible to achieve better image detail and lower dose exposure with xeroradiography, which proved to be an excellent alternative to the old X-ray units. These results were later confirmed in the late 1960s by studies of Dr. Wolf JN and Dr. Martin GE, who demonstrated that the inherent contrast of this technology was well adapted to the study of the breast [18]. The advent of the xeromammography technique and the development of screen-film mammography [14,19] encouraged researchers to investigate ever more precise systems fully dedicated to the study of the breast. Then, in the late 1960s, Charles Gros developed the first X-ray device designed exclusively for mammograms using a molybdenum tube instead of the tungsten tube. Professor Gros also emphasized the potential of mammography in the detection of occult cancer, laying the foundation for subsequent screening trials [20,21]. Along with Professor Gros, another researcher who contributed to the clinical application of technical developments was Robert Egan (1921–2001) [22,23]. During his career, he enthusiastically championed mammography in the United States, training radiologists and technicians and leading to a study by the Cancer Control Program (CCP) of the U.S. Public Health Service, which involved 24 institutions and 1500 patients. His studies suggested that mammography could be used for screening because it could find tumors that could not be seen through physical examination and identify lesions eligible for biopsy, even if not clinically palpable. Until then, mammography was only used in more complicated cases, where the physical examination was inconclusive.

Moreover, the moving grid for mammography was introduced in 1978. This technique reduces scattered radiation and further improves intrinsic image contrast [24]. This further technological innovation allowed companies to develop better equipment for mammography, capable of producing more detailed images.

The requirement for images with high spatial resolution was confronted with the need not to increase the dose absorbed by patients. Using dedicated complex physical models, such as the Monte Carlo model, introduced in the 1980s [25], this led to the development of anti-scatter grid technology that increased spatial resolution while reducing absorbed dose. In the last decade, virtual grid technology software, based on mathematical algorithms, has been developed to reduce exposure dose and image artifacts further and provide an image with more contrast [26].

Between 1986 and 1992, the U.S. federal and state governments enacted laws that forced hospitals to use dedicated machines for mammography, that set quality standards, and that required periodical inspections of equipment [27]. The addition of a mediolateral oblique view as a standard mammographic projection, as well as the introduction of automated compression systems of the breast, allowed better visualization of the mammary gland and its structure [27]. All these technological developments led to the introduction of digital mammography, approved by the Food and Drug Administration in 2000 [28]. In digital mammography, the transmitted radiation is transferred on an electronic image detector instead of on film. The main advantage of this is to decouple image acquisition and image storage, with the principle behind digital mammography being to obtain as much information as possible with the lowest dose. In the history of mammography detectors, the leading technological development with the advent of digital technology has been to free the detector from storing images and focus it on absorbing X-rays and producing high-energy images while reducing the dose absorbed by patients. The detector produces a signal based on the absorbed energy, which is then transformed into an image.

Full-field digital mammography (FFDM) was proved to have significantly higher diagnostic accuracy in the identification of breast lesions when compared with screen-film mammography [29].

A typical example of a full-field digital mammograph is shown in Figure 1.

## 4. Mammography Screening and Reduction in Breast Cancer Mortality

In the 1970s, breast cancer was one of the leading causes of mortality in the United States [30]. The possibility of finding interventions that could address this situation was, therefore, indispensable. Philip Strax (1909–1999), a radiologist from New York, stressed the need to study women with an entirely negative physical examination. The discovery of non-palpable cancers and their identification before distal spreading has opened up new scenarios in managing breast neoplasia [31]. Philip Strax, in collaboration with the private insurance company Health Insurance Plan of Greater New York (HIP), published the first significant results on the reduction in mortality produced by screening. This research group performed a study that included sixty thousand women aged forty to sixty-four [32]. Half of the women received an annual clinical breast examination that included mammography, while the rest were not invited for an annual follow-up. After four years, in 1971, the HIP investigators published an article in the Journal of the American Medical Association [32]. The researchers found that women in the intervention group had a mortality rate thirty percent lower than those who had not been followed up. It was the first time such a large-scale study had demonstrated that mammography could lower the mortality rate of breast cancer. In this study, the first one on breast screening, the attendance rate was not very high, at around 46%. These findings led other researchers to initiate other studies on screening that were published in the following years. In 1988, the first results of the Malmo screening trial were published [33]. This study was conducted on women aged 43 to 69 years with 21,088 women being invited to screening; 21,195 women were in the control group. A reduction in mortality of more than 20% was demonstrated in women aged 55 years and older. The attendance rate of this study was better than the previous study, with adherence rates around 70%. Moreover, in 1989 and 1991, the results of the Swedish two-country trial and the Stockholm trial were published [34,35]. These studies on thousands of people also confirmed a significant reduction in breast cancer mortality (greater than 30%), with mammography screening adherence rates this time exceeding 80 percent. Two other essential studies we intend to mention are The Gothenburg screening trial published in 1997 [36] and the UK screening trial [37]. These two studies are critical because, in addition to confirming the results on overall mortality reduction, they also demonstrate mortality reduction in women under 50. Based on the results of these studies, some researchers have concluded that in women under 50, yearly screening may be suggested with a more significant effect on mortality [38]. In addition, given the excellent results of these screening protocols, mammography in the 1980s underwent exceptional development [39]. These crucial achievements led to the need to categorize breast lesions and the creation of The Breast Imaging Reporting and Data System (BI-RADS) in 1993, which has now reached its fifth edition. The advent of BI-RADS made it possible to overcome the lack of consistency in the interpretation of mammography findings and improved communication among breast-lesion handlers. Breast imaging findings are divided into seven categories, from 0 to 6, where BI-RADS category 1 indicates the absence of abnormal findings and BI-RADS category 6 represents a biopsy-proven malignancy; moving up in the scale, the level of suspected malignancy progressively increases, while 0 means an inconclusive mammogram. Since its introduction, the BI-RADS lexicon has proven to be an effective way for radiologists to evaluate imaging findings and to decide if additional procedures are required [40,41].

The main problems still unresolved with screening mammography are the low sensitivity of mammography in dense breasts [42] and the high number of recalls for false-positive findings [43]. To a lesser extent, another problem facing screening is overdiagnosis: the diagnosis of indolent breast neoplasms that would not have become clinically evident during the patient’s lifetime [44]. Efforts have been made to cope with these issues by developing other breast imaging methods whose history is mentioned later in our paper.

## 5. Digital Breast Tomosynthesis (DBT)

Tissue overlap, especially in dense breasts, may result in difficulty in the detection of some breast lesions [45]. Digital breast tomosynthesis was developed to overcome this issue. In tomosynthesis, multiple projections are acquired through a predefined trajectory to obtain three-dimensional and sectional images of the breast. These sectional images attempt to overcome the limitation of tissue overlap. The first generic concept of tomosynthesis had already been mentioned in the 1930s [46]. However, clinical application in the breast was first hypothesized in the 1990s, thanks in particular to the studies of Dr. Niklason and Dr. Daniel B Kopans, head of the Xeroradiography Division at Massachusetts General Hospital [47].

This new development sought to address the reduced sensitivity of conventional mammography in identifying malignancies in dense breasts [48].

By producing three-dimensional (3D) images of the breast, this technique is able to reduce anatomic noise or overlap of glandular tissue, improving the visibility of nodules and architectural distortions of the mammary gland [49].

Some studies have introduced the use of tomosynthesis in a screening context with encouraging results. One of the most important prospective studies that has evaluated the role of tomosynthesis in screening is the OTST (Oslo Tomosynthesis Screening Trial) [50,51]. In this study, in a group of 12,631 examinations, an increased cancer detection rate from 6.1 per 1000 examinations to 8.0 per 1000 examinations and a reduction in the false-positive rate from 10.3 percent to 8.5 percent were shown by tomosynthesis. Compared with full-field digital mammography analysis alone, 27 more invasive carcinomas were detected. In the Storm trial [52]; a study based on the analysis of 7292 examinations was found to have a 17% reduction in false-positive rate and an increase in cancer detection rate from 5.3 per 1000 to 8.1 per 1000.

Numerous subsequent studies have confirmed these encouraging results [53,54,55,56,57,58].

Some of the problems still unresolved from using tomosynthesis are the significant additional radiation exposure (compared to full-field digital mammography) [59] and the controversial ability of tomosynthesis acquisitions to identify and correctly classify microcalcifications [60]. Finally, the visibility (detection) of lesions in very dense breasts remains controversial [61].

## 6. Contrast-Enhanced Mammography (CEM)

In the 1930s and 1940s, Emil Ries and Nymphus Frederick Hicken (1900–1998) investigated contrast-enhanced mammography to obtain better visualization of the mammary ducts [62]. However, the foundations for the development of CEM were laid in the 1970s when preliminary studies showed that intravenously injected iodinated medium contrast was associated with an enhancement of breast cancer lesions on computed tomography (CT) [63,64]. Later, in the 1980s, the concept of dual-energy subtraction (a concept based on time subtraction to improve contrast visibility) was first developed by William Brody and colleagues [65].

However, the first published article on the clinical experience of using CEM was published by Jong et al. in 2003 [66]—for the first time, this study demonstrated the good ability of contrast applied to mammography to identify breast lesions.

After some preliminary studies confirmed these encouraging results, General Electric invested in producing a prototype specifically for performing dual-energy subtraction mammography examinations [67,68].

The first paper that used this specific mammograph was published in 2011 by Dromain et al. [69].

The performance of CEM in identifying breast lesions was significantly better than mammography alone. CEM received approval from the Food and Drug Administration in the same year.

CEM is based on dual-energy breast exposure (about 26–33 kVp and 44–50 kVp) after contrast administration and allows visualization of breast lesions that take contrast. The examination is usually completed within 5 min of contrast medium injection. The most common acquisition technique is to perform a craniocaudal and mid-lateral oblique projection per side, injecting a 1.5 mL/kg contrast dose with a 3 mL/s flow rate.

The average glandular dose remains within limits set by the European guidelines for screening mammography [70].

The most recent studies demonstrate that CEM has the potential for exceptional diagnostic performance [71] as it can combine the advantages of mammography with those of breast magnetic resonance. CEM, as well as MRI, seems to contribute to the diagnosis of tumors that are difficult to detect with conventional mammography and, sometimes, digital breast tomosynthesis (especially in dense and extremely dense breasts) with some important advantages compared to MRI: contrast-enhanced mammography is easier and faster to perform, less expensive, and has a less psychological impact on patients. It also appears to produce fewer false-positive results and is also feasible in patients for whom MRI cannot be performed [72] (e.g., those with pacemakers).

The potential of CEM in clinical practice as a second-level method is more than encouraging. For example, CEM seems to have an essential role in the correct definition of the extent of a lesion, before surgery, as a safe support for conservative surgery [73]. CEM also offers a significant added value in evaluating breast microcalcifications, being able to avoid useless biopsies for benign lesions [74] and, in general, it offers excellent diagnostic performance in the prediction of malignancy of breast lesions providing a good ratio between the number of false positives and false negatives [75]. Excellent results have also been demonstrated in its use in evaluating suspicious findings encountered with organized mammography screening in Europe where CEM seems to reduce the number of recalls in screening for benign lesions [76]. Despite these crucial results, its role as a potential substitute for screening mammography in some patient populations (e.g., high-risk patients with dense breasts) is only just beginning [77]. CEM also seems helpful in neoadjuvant chemotherapy response monitoring [78]. With these findings, CEM would seem to be the most recent evolution of mammography, providing clinical implications of great interest in the coming years and a supplement to the mammography BI-RADS was recently published to standardize the findings that can be detected by CEM examination among radiologists [79].

A typical example of a lesion visible only with a CEM recombined image (after injection of contrast medium) is seen in Figure 2.

In Figure 2a, we can appreciate the craniocaudal acquisition of conventional mammography. In Figure 2b, we can see the acquisition of the recombined image of the CEM after the injection of the contrast medium. Only in Figure 2b can we see a breast lesion, with slight enhancement, in the outer quadrants (arrow). The subsequent biopsy showed a papilloma (B3 lesion).

In Figure 3, we have described some of the main achievements in the development of mammography.

## 7. Breast MRI

Early research into the use of MRI in the breast began around the 1980s when Dr. Heywang demonstrated that breast neoplasia took contrast to the surrounding parenchyma after the intravenous injection of contrast medium [80]. Dr. Heywang’s studies were later implemented by the group of Kayser et al., who studied protocols that could achieve better spatial and temporal resolution in breast MRI [81]. In subsequent years and particularly in the 1990s, researchers have focused on the possibility of further improving techniques for acquiring and interpreting MRI-acquired findings. The results obtained by Dr. Kuhl’s research group are of particular note in this area. These studies have mainly focused on the ability to find a trade-off between spatial and temporal resolution [82]. The purpose was to obtain information that was as optimized as possible. After these results, and up to the present day, resonances are acquired with 1.5 or 3 T coils with high spatial and temporal resolution. Today, breast MRI is indispensable in the management of patients with breast lesions. Among the leading applications whose usefulness is established, we find the proper assessment of the extent of lesions to evaluate conservative surgery, evaluation of response to neoadjuvant therapy, study in patients with axillary adenopathy without finding mammographically evident breast lesions, and study of previous surgery scars and prosthetic integrity evaluation [83]. However, one of the most exciting roles to be discussed is the potential of MRI to be used as an additional screening method in high-risk patients. The first screening study, performed on high-risk women, again by the group of Kuhl et al. on 192 patients, showed the findings of six neoplasms that were not visible on either ultrasound or mammography [84]. A subsequent study conducted in a large population confirmed these more than encouraging and superior results compared with mammography screening [85].

The most recent published meta-analysis confirms that MRI is the best additional imaging modality in women at highor intermediate risk for breast cancer with dense breasts and negative mammography for cancer [86].

Breast MRI also has shortcomings, e.g., long acquisition time, high cost, a high number of false positives, and inability to perform in all patients (e.g., claustrophobia, pacemakers) [87]. In some cases, these shortcomings could be well compensated by the promising properties of CEM, which still need to be tested extensively in screening settings with high-risk patients.

## 8. Breast Ultrasound

High-frequency ultrasound for the study of the breast was first described by Dr. Wild in 1951 [88]. Early research was not focused on using ultrasonography as a method of tumor detection but on distinguishing the characteristics of benign and malignant breast disease [89].

The development of increasingly accurate equipment characterized the next 20 years. In particular, ultrasound probes with improved crystals and ultrasound beam collimation systems and techniques based on different image gray scales were developed [90]. These technological advances were later consolidated in the 1980s with the advent of digital technology and the development of color Doppler, a technique capable of assessing the presence of vascularity within a breast lesion [91]. In the late 1980s, the importance of breast ultrasonography as a guide for biopsy procedures and in preoperative assessment of the size of a breast lesion was established [92,93]. Breast ultrasound makes precise and detailed pre-surgical anatomical planning available, especially with the most modern image-acquisition techniques, such as the lobar approach [94].

It is known that ultrasound, complementary to mammography, in a breast with a high glandular component, is excellent in improving tumor detection. However, some aspects make its use as a screening method controversial: ultrasound is very operator dependent, difficult to standardize, and “time-consuming”. In addition, as a screening method, it increases the number of false positives [95].

In recent years, the role of automated breast ultrasound (ABUS) has become increasingly promising. This new technology mitigates some of the main shortcomings of conventional manual ultrasound, although its use has yet to be fully established in clinical practice [96].

## 9. Future Directions

The last 100 years have seen astonishing developments in imaging applied to breast malignancies; however, we believe that efforts should not stop yet. Scientists must find the most innovative personalized approaches that provide all patients with the best diagnostic and therapeutic options [97]. Particularly in recent years, more consideration has been given to providing patients with personalized diagnostic and screening approaches, especially to the latest diagnostic developments such as CEM and MRI [97].

The last few years have also seen increasing development of artificial intelligence based on computer algorithms capable of emulating human behavior and of radiomics, which can be defined as the ‘genetics of the image’—images possess specific intrinsic characteristics that can be extracted and correlated (with therefore predictive power) with complex mathematical algorithms to specific clinical outcomes.

Artificial intelligence and radiomics fit well into the context of precision and personalized medicine, capable of optimizing patients’ diagnostic and therapeutic workflow.

For instance, artificial intelligence could better assist in detecting and interpreting small breast lesions by reducing the number of false-positive and false-negative interpretations of images [98,99]. Moreover, radiomics seems to predict crucial prognostic information inherent in the image [100]. With this information, managing high-risk patients could be different and personalized and the role of breast imaging in cancer screening and diagnosis may continue to evolve in the coming years [101].

Among the most recent applications of radiomics applied to mammography, we have, for example, the possibility of better predicting the benignity or malignancy of breast lesions, with consequent greater optimization in the requests for breast biopsies [102], the possibility of predicting the receptor status of a breast neoplasm before biopsy [103], and the possibility of predicting the underestimation of a breast biopsy [104].

These rapid developments have to face some challenges and issues, such as the ethical and legal implications of bias and standardization, privacy, and cybersecurity [105]. In the coming years, it will therefore be necessary to find a compromise between the progress of these aspects and the ethical and data-privacy issues.

Finally, the latest technological advances should focus on providing the patient with diagnostic methods that can increasingly reduce the discomfort associated with the examination. The diagnostic study should focus on the person and thus consider the possible pain resulting from a harrowing investigation, exceptionally one that is long or performed in particularly uncomfortable positions. Indeed, the latest generation of diagnostic equipment also fulfils this aspect, and it will have to be given increasing attention in the years to come. Diagnostic imaging should become increasingly patient-driven, considering all aspects of the ‘person,’ not just the patient [106].

## 10. Conclusions

In nearly 100 years, mammography and breast imaging progress and potential applications have been incredible and dizzying. In this review, we have provided an overview of the historical evolution of breast imaging over the years. A deeper knowledge of the history of the evolution of breast imaging can lead to facing future changes with greater awareness and with a constant goal: the best possible personalized management in the patient with breast neoplasia.

## Figures and Tables

**Figure 1 healthcare-11-01596-f001:**
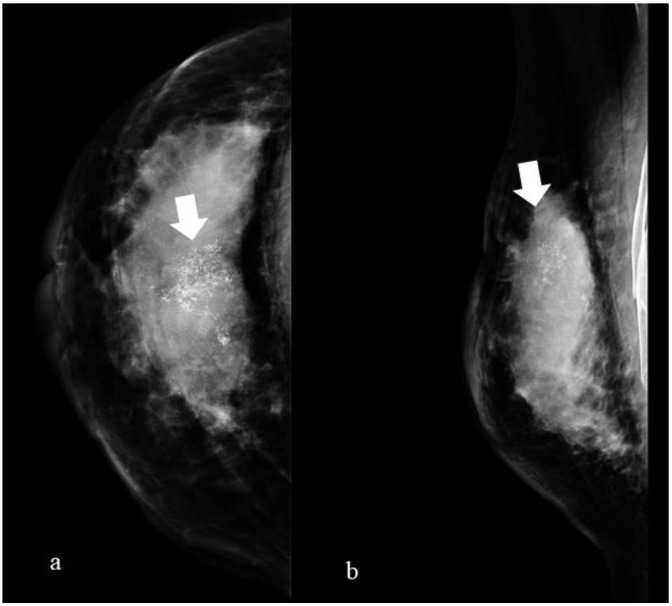
A 45-year-old woman with infiltrating ductal carcinoma of the right breast. In the figure, we can observe a craniocaudal (**a**) and mediolateral oblique projection (**b**) radiogram of the right breast with the presence of polymorphic microcalcifications extended to the upper sectors (arrow).

**Figure 2 healthcare-11-01596-f002:**
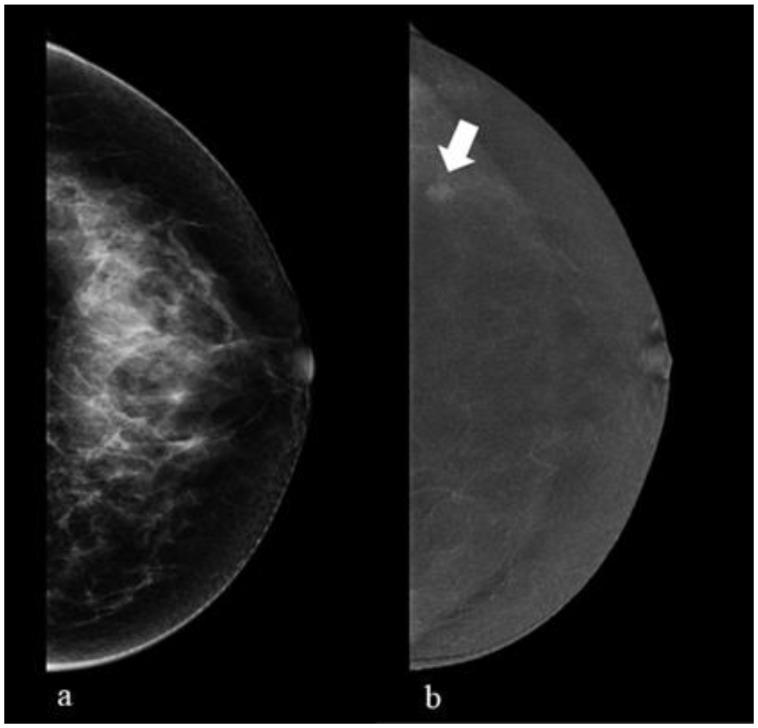
A CEM recombined image (after injection of contrast medium) from a 48-year-old woman with dense breasts.

**Figure 3 healthcare-11-01596-f003:**
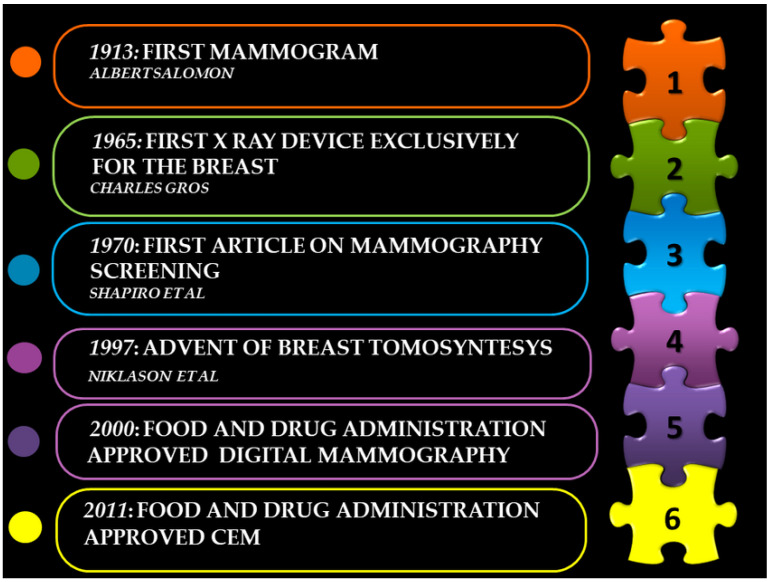
Some of the main achievements of mammography over time.

## Data Availability

Not applicable.

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
