# Peer review of "History of Mammography: Analysis of Breast Imaging Diagnostic Achievements over the Last Century"

_healthcare, 2023, doi:10.3390/healthcare11111596_

Round 1
Reviewer 1 Report (Previous Reviewer 2)
History of mammography: analysis of breast imaging diagnostic achievements over the last century
In this study the authors conducted a historical review of the development of mammography and breast imaging over the last century. The review aims to offer a complete panorama of the development of mammography and breast imaging, from the foundations of radiology applied to the breast to the most modern applications such as Contrast enhanced mammography (CEM), artificial intelligence, and radiomics. The goal of the review is to understand where the foundations of the pillar of radiology applied to the breast start-up and how to direct efforts toward an increasingly personalized and effective diagnostic approach. The ultimate goal is to reduce mortality from breast cancer maximally and offer the best prognosis and quality of life for patients. The paper also discusses possible current and future applications of imaging aimed at being more precise and personalized.
Study design: Historical / narrative review.
General comments
This is my second revision of this manuscript.
The intention of the review was to provide a comprehensive overview of the development of mammography, from its early beginnings to modern applications. I believe that this historical or narrative review does not add any value to what is known about breast cancer diagnostic techniques. The second revision is a significant improvement from the first one. I am grateful that additional sections were added as requested, and the figure was included. Furthermore, I appreciate that the references have been updated to incorporate the most current evidence.
The content appears to be informative and well-researched, with appropriate references to support the claims made.
There are two more points I'd like to raise regarding this work:
1) I suggest enhancing Figure 1 to make it more visually appealing and eye-catching. While the current version created with PowerPoint's SmartArt function is good, adding images or figures that represent the iconographic evolution of breast imaging could make it even better. Perhaps imagining it more as a “visual abstract” would be helpful.
2) The article could benefit from more references to the patient and their comfort during the examination. Over time, there has been a focus on improving aspects related to the patient, such as reducing radiation exposure and making the exam more comfortable. While you did mention the psychological impact of CEM vs. MRI (“has a less psychological impact on patients” line 270), I think it would be useful to emphasize these points in the conclusion or final part of the review. It's important for new and future technologies to take into account psychological aspects and become increasingly patient-driven, considering all aspects of the “person”, not just the patient.
I did not detect any critical issues in the grammar or syntax of the article. There are a few instances of sentence fragments, but these appear to be intentional for stylistic reasons. Overall, the article is well-written and readable.
Author Response
This is my second revision of this manuscript.
The intention of the review was to provide a comprehensive overview of the development of mammography, from its early beginnings to modern applications. I believe that this historical or narrative review does not add any value to what is known about breast cancer diagnostic techniques. The second revision is a significant improvement from the first one. I am grateful that additional sections were added as requested, and the figure was included. Furthermore, I appreciate that the references have been updated to incorporate the most current evidence.
The content appears to be informative and well-researched, with appropriate references to support the claims made.
- Thanks for your comment that gratify our work.
There are two more points I'd like to raise regarding this work:
I suggest enhancing Figure 1 to make it more visually appealing and eye-catching. While the current version created with PowerPoint's SmartArt function is good, adding images or figures that represent the iconographic evolution of breast imaging could make it even better. Perhaps imagining it more as a “visual abstract” would be helpful.
- The figure has been modified as per your suggestion. We think it is now much improved and similar to a visual abstract
2) The article could benefit from more references to the patient and their comfort during the examination. Over time, there has been a focus on improving aspects related to the patient, such as reducing radiation exposure and making the exam more comfortable. While you did mention the psychological impact of CEM vs. MRI (“has a less psychological impact on patients” line 270), I think it would be useful to emphasize these points in the conclusion or final part of the review. It's important for new and future technologies to take into account psychological aspects and become increasingly patient-driven, considering all aspects of the “person”, not just the patient.
- We thank you for this comment and agree with you. As you suggested, we have emphasized this concept in the final part of the review and have taken the liberty of including in the text some of your words that we particularly liked
Comments on the Quality of English Language
I did not detect any critical issues in the grammar or syntax of the article. There are a few instances of sentence fragments, but these appear to be intentional for stylistic reasons. Overall, the article is well-written and readable.
Reviewer 2 Report (Previous Reviewer 4)
The revised manuscript has become much longer compared to the previous version. The number of cited references has doubled, some sections has become widened partly with unnecessary overhelming information on the the number of included patients and other detailes in certain studies. This was probably done after suggestions of the other reviewers; I personally liked the previous version much more.
Nevertheless, the actual version is also good enough for publication (I would skip the unnecessary study details mentioned above), but there is a problem. Namely, if MRI is discussed in such details, breast ultrasound must also discussed in the text. I do not see any reason to negligate this modality as it has crutial role in breast imaging. It could be ommitted in a shorter review focused on the history of mammography (previous version of the manuscript), but not in the actual detailed version. And to achieve a balans, a paragraph about ultrasound should be approximately as long as the MRI section, at least.
The section "Future directions" seems to be a "rush job" with repetitions and disturbed logical order. Please reformulate it.
The text is well-written, but there are a few mistakes.
- Line 36: maximally and best: this means 0 mortality, which is not achievable. I would use "as much as possible" or "to an achievable minimum". It means at the same time "best prognosis". (I mean "...to an achieveble minimum as well as emproving the quality of life...")
- lines 121, 123: Gros is not the only professor cited in the text, why only him is cited with this title?
- Line 331 "long" acquisition time, not "high".
Author Response
The revised manuscript has become much longer compared to the previous version. The number of cited references has doubled, some sections has become widened partly with unnecessary overwhelming information on the number of included patients and other details in certain studies. This was probably done after suggestions of the other reviewers; I personally liked the previous version much more.
-We thank you for the comment. Unfortunately, this radical change in the paper was necessary as a result of other reviewers' comments.
- It has cost us much work to make this change, and we have tried to balance the attempt to be concise with comprehensiveness in reporting all the results of our research.
Nevertheless, the actual version is also good enough for publication (I would skip the unnecessary study details mentioned above), but there is a problem. Namely, if MRI is discussed in such details, breast ultrasound must also discussed in the text. I do not see any reason to negligate this modality as it has crutial role in breast imaging. It could be ommitted in a shorter review focused on the history of mammography (previous version of the manuscript), but not in the actual detailed version. And to achieve a balans, a paragraph about ultrasound should be approximately as long as the MRI section, at least.
- We thank you for the comment and suggestion.
We have also introduced a paragraph on the history of ultrasound in the text.
The section "Future directions" seems to be a "rush job" with repetitions and disturbed logical order. Please reformulate it.
- We thank you for the suggestion.
We have changed the structure of the last paragraph by removing the repetitions in particular.
Comments on the Quality of English Language
The text is well-written, but there are a few mistakes.
- Line 36: maximally and best: this means 0 mortality, which is not achievable. I would use "as much as possible" or "to an achievable minimum". It means at the same time "best prognosis". (I mean "...to an achieveble minimum as well as emproving the quality of life...")
- Thanks, we changed the text according to your suggestion.
- lines 121, 123: Gros is not the only professor cited in the text, why only him is cited with this title?
- Thank you for the clarification. As a matter of synthesis, we reported what we felt was the most important author. However, in the line just below, Dr Egan is also mentioned.
- Line 331 "long" acquisition time, not "high".
Thanks, we changed the text according to your suggestion.
Reviewer 3 Report (New Reviewer)
My comments are attached as a word fil. Please find the attached document.

Author Response
Type: Review Title: History of Mammography: analysis of breast imaging diagnostic achievements over the last century Author: Luca Nicosia et al.
General comments: This manuscript contains a lot of good and relevant information and mostly covers the relevant historical development of mammography. However, I have some short specific comments to improve the quality of the manuscript.
- We thank you for the positive comment which gratifies our work
- In line 50 the author mentioned “mammography is low dose X-ray technique”, which is not sound correct in my knowledge of X-ray imaging systems mammography is relatively dose-intensive compared to other X-ray modalities. Ideally, mammography uses a much lower tube energy (kVp) and a higher mAs value, resulting in a higher radiation dose to patients than conventional X-rays, for example. Therefore, this term is somewhat odd unless the author adds further clarification by reference.
-we thank you for the suggestion.
-We have eliminated the term low dose which risked being incorrect.
- The authors have done a very thorough study of the development of clinical mammography and a bit about the technological development of mammography. However, I think they did not mention the development of the image detector. In addition, how the detector has evolved since the discovery of mammography. Because the detector and its development are of great importance for the detection of cancer, and the distinguishing of micro-calcifications and malignant changes in the breast. The detector can also play an important role in terms of radiation dose to patients. Therefore, I would suggest that the authors discuss a little about how image detection in mammography has developed over the last century.
- Thank you for the comment.
-We have implemented some details on the development of detectors in mammography as per your request.
-We have remained brief because the main purpose of our research is clinical: therefore, we have tried to report the details that we felt were most related to changes in clinical practice.
- The authors mentioned also the physical grid in mammography and its role in imaging improvement. However, the physical inti-scatter grid has also limitations including contributing to higher radiation exposure to the patient. Over time, a lot has happened in the field of anti-scatter grid technology, virtual grids have been on the market for ten years. I think you should also mention this point, which is important both for image quality and for the radiation dose to patients.
-Thank you for the comment.
-We have implemented some details on the development of anti-scatter grid technology as per your request.
-We have remained brief because the main purpose of our research is clinical: therefore, we have tried to report the details that we felt were most related to changes in clinical practice.
- The citation format is not consistent. For example, some citations are rendered as "Jong et al 2003". In some other cases, and mostly by the authors, the titles of the studies are given followed by the authors and year. I would suggest that authors find the alignment of citations throughout the manuscript.
Thank you we have corrected the style of the citations.
- The resolution of Figure 3 is too low. The text in the blue box is too small. This must be improved.
-Thank you Figure 3 has been changed completely (also related to another reviewer's comments). The text is larger and the resolution increased.
- It is good to discuss MRI in breast imaging. However, there are other important breast imaging modalities, including breast ultrasound and breast CT, which are not discussed in great detail in the manuscript. It would be nice if the authors would cover these mammography techniques a little.
-We thank you for the comment. We have added a paragraph on ultrasound (with mentions of automated ultrasound as well). In our clinical experience (we are dedicated breast radiologists in an oncology referral hospital), dedicated breast CT is not mainly used. Therefore, we prefer not to add paragraphs on this method (also because the article would risk becoming very long and scatter relevant information).
Round 2
Reviewer 2 Report (Previous Reviewer 4)
This revision of the manuscript has becomew a comprehensive review of the history of breast imaging. It is now acceptable for publication.
I still have one suggestion regarding the paragraph about breast ultrasound: Lobar approach in breast ultrasound / ductal echography of Michel Teboul and Dominique Amy could be mentioned as it has followers in many countries. This could be a minor change for the completness of the great work done.
Author Response
This revision of the manuscript has become a comprehensive review of the history of breast imaging. It is now acceptable for publication.
I still have one suggestion regarding the paragraph about breast ultrasound: Lobar approach in breast ultrasound / ductal echography of Michel Teboul and Dominique Amy could be mentioned as it has followers in many countries. This could be a minor change for the completeness of the great work done.
Thanks for the comment which gratifies our work.
We have cited and briefly discussed the work you suggest.
This manuscript is a resubmission of an earlier submission. The following is a list of the peer review reports and author responses from that submission.
Round 1
Reviewer 1 Report
The type of paper is opinion. I was expecting that the authors discuss a specific issue in mammography other than briefly summarizing the history of mammography. For this reason, I will not consider this paper an opinion paper.
The paper is more like a review paper; however, it falls short in reviewing the history of mammography, its advantages, and shortcomings in detection and diagnosis. The paper is well written, but it is a very surface level that its contribution to the field is low.
The paper's title gives the impression that the future of mammography is discussed. However, it did not go beyond Contrast Enhanced Mammography, which was approved in 2011.
Machine learning-based computer-aided detection or diagnosis systems are applications that use mammography images. These should not be considered the future of mammography technology.
I ask the authors these questions:
What is the future of breast mammography?
What are the advancements in medical imaging technologies to make mammography more precise, affordable, easier to use, smaller in size, etc.?
Author Response
The type of paper is opinion. I was expecting that the authors discuss a specific issue in mammography other than briefly summarizing the history of mammography. For this reason, I will not consider this paper an opinion paper.
The paper is more like a review paper; however, it falls short in reviewing the history of mammography, its advantages, and shortcomings in detection and diagnosis. The paper is well written, but it is a very surface level that its contribution to the field is low.
The paper's title gives the impression that the future of mammography is discussed. However, it did not go beyond Contrast Enhanced Mammography, which was approved in 2011.
Machine learning-based computer-aided detection or diagnosis systems are applications that use mammography images. These should not be considered the future of mammography technology.
I ask the authors these questions:
What is the future of breast mammography?
What are the advancements in medical imaging technologies to make mammography more precise, affordable, easier to use, smaller in size, etc.?
We thank you for the review.
The article has been thoroughly revised: we think we have addressed the concepts more deeply and, per your suggestion, have framed a possible future development of breast imaging. As per your suggestion, we have removed the mention of computed aided detection imaging
We hope you will positively re-evaluate our work.
Finally, the article has been submitted as a review, as per your suggestion, and not as an opinion paper.
Reviewer 2 Report
History of breast mammography: the study of the beginning to understand where we are traveling to
In this study the authors conducted a historical review on the development of diagnostic imaging techniques for breast cancer.
Study design: Historical / narrative review.
General comments
This is my first revision of this manuscript.
The intention of the review was to provide a comprehensive overview of the development of mammography, from its early beginnings to modern applications. I believe that this historical or narrative review does not add any value to what is known about breast cancer diagnostic techniques; however, it is an interesting read and is well written, especially in the first part. I think it could be more expanded in the second part providing a perspective of what will be the future of breast imaging.
Introduction
Please better introduce the aim and the context of the review, and in the second part please try to better discuss the evolution of acquisition techniques and what the future holds for breast diagnostics. In addition to these techniques, other developments in the field of breast diagnostics include the use of molecular imaging and functional imaging, such as positron emission mammography (PEM) and magnetic resonance imaging (MRI). These techniques offer improved sensitivity and specificity in detecting breast cancer and may complement mammography in the future.
History of mammography
Very interesting paragraph, please add a figure that can represent a timeline of the evolution of mammography technique, for example, with the indication of these key dates or those that you yourself have already used in the text:
1913: The first radiograph of a breast lesion was taken using X-rays.
1950s: The advent of the xeromammography technique, which uses a dry photographic process instead of X-ray film.
1960s: The development of screen-film mammography, which became the standard of care for breast cancer screening.
1970s: Introduction of stereotactic biopsy, a minimally invasive biopsy technique that uses mammography to guide the needle into the breast tissue.
1990s: Introduction of digital mammography, which uses digital detectors instead of X-ray film.
2000s: The development of breast tomosynthesis, a 3D mammography technique that offers improved accuracy in breast cancer detection.
2010s: Introduction of contrast-enhanced mammography, a technique that uses contrast agents to highlight blood flow in breast tissue and improve the accuracy of breast cancer detection.
Through a simple figure, you can quickly describe a brief overview of the evolution of mammography techniques over the last century, from the first radiograph of a breast lesion to modern techniques such as contrast-enhanced mammography.
Contrast-enhanced mammography paragraph
The paragraph on contrast enhanced mammography is quite rough, please add the most up-to-date literature references, both those concerning the acquisition technique (please see: 10.1186/s13244-019-0756-0) and the latest in terms of diagnostic performance (please see: 10.1148/radiol.211412).
Please be consistent with the terminology used, I agree to use the acronym CEM and no longer the acronym CESM as you mistakenly use on page 5 line 196
The final part may further discuss the impact of targeted therapies and precision medicine on mammography screening (please see: 10.1186/s13244-020-00905-3). While targeted therapies and precision medicine have improved the treatment of breast cancer, early detection through screening remains crucial for improving outcomes and reducing mortality.
Also, in my opinion it is really necessary to add a small paragraph on the role of the breast MRI to predict response to neoadjuvant chemotherapy and how can be used as a prognostic tool. Another area of research is the use of breast MRI for risk stratification in women with high genetic risk of breast cancer. As you know, breast MRI has great potential for predicting and prognosticating breast cancer (please see: 10.1186/s41747-022-00291-z).
Conclusion
In my opinion, you can still elaborate a few more steps and clarify that with the advent of targeted therapies and precision medicine, the role of mammography in breast cancer screening and diagnosis may continue to evolve in the coming years. In any case, as you have amply described, mammography remains a critical tool in the early detection of breast cancer, and ongoing research efforts are focused on improving its performance and expanding its potential applications.
Author Response
History of breast mammography: the study of the beginning to understand where we are traveling to
In this study the authors conducted a historical review on the development of diagnostic imaging techniques for breast cancer.
Study design: Historical / narrative review.
- General comments
This is my first revision of this manuscript.
The intention of the review was to provide a comprehensive overview of the development of mammography, from its early beginnings to modern applications. I believe that this historical or narrative review does not add any value to what is known about breast cancer diagnostic techniques; however, it is an interesting read and is well written, especially in the first part. I think it could be more expanded in the second part providing a perspective of what will be the future of breast imaging.
We thank you for the well-done revision that helped improve our work: the article has been thoroughly revised according to your comments and the comments of the other reviewers. In particular, we have tried to better develop the second part according to your suggestions.
- Introduction
Please better introduce the aim and the context of the review, and in the second part please try to better discuss the evolution of acquisition techniques and what the future holds for breast diagnostics. In addition to these techniques, other developments in the field of breast diagnostics include the use of molecular imaging and functional imaging, such as positron emission mammography (PEM) and magnetic resonance imaging (MRI). These techniques offer improved sensitivity and specificity in detecting breast cancer and may complement mammography in the future.
We have better specify the objective of our work. We have better developed the paragraph on CEM and introduced a paragraph on Breast MRI.
We have highlighted the clinical applications of these methods. In particular, throughout the article, we have tried to focus on the possibility of such methods being introduced in a particular category of patients in a screening context in the context of personalized medicine.
Unless you consider it indispensable, we have decided not to consider the use of PEM (positron emission mammography) in our work. The main disadvantage of PEM is the radiation exposure: single PEM study involving the use of a label-recommended radionuclide dose is associated with a 15-fold higher risk of cancer induction than a single screen film or digital mammogram (Glass SB, Shah ZA. Clinical utility of positron emission mammography. Proc (Bayl Univ Med Cent). 2013 Jul;26(3):314-9. doi: 10.1080/08998280.2013.11928996. PMID: 23814402; PMCID: PMC3684309). We believe that this contraindicates its use in daily clinical practice and prevents future clinical applications compared to methods such as CEM and MRI
- History of mammography
Very interesting paragraph, please add a figure that can represent a timeline of the evolution of mammography technique, for example, with the indication of these key dates or those that you yourself have already used in the text:
1913: The first radiograph of a breast lesion was taken using X-rays.
1950s: The advent of the xeromammography technique, which uses a dry photographic process instead of X-ray film.
1960s: The development of screen-film mammography, which became the standard of care for breast cancer screening.
1970s: Introduction of stereotactic biopsy, a minimally invasive biopsy technique that uses mammography to guide the needle into the breast tissue.
1990s: Introduction of digital mammography, which uses digital detectors instead of X-ray film.
2000s: The development of breast tomosynthesis, a 3D mammography technique that offers improved accuracy in breast cancer detection.
2010s: Introduction of contrast-enhanced mammography, a technique that uses contrast agents to highlight blood flow in breast tissue and improve the accuracy of breast cancer detection.
Through a simple figure, you can quickly describe a brief overview of the evolution of mammography techniques over the last century, from the first radiograph of a breast lesion to modern techniques such as contrast-enhanced mammography.
We have added a figure according to your comment. (Figure 3) We have tried to include in the figure what, among the many described, seem (in our opinion) to be the most important steps in the development of the mammography story.
- Contrast-enhanced mammography paragraph
The paragraph on contrast enhanced mammography is quite rough, please add the most up-to-date literature references, both those concerning the acquisition technique (please see: 10.1186/s13244-019-0756-0) and the latest in terms of diagnostic performance (please see: 10.1148/radiol.211412).
Please be consistent with the terminology used, I agree to use the acronym CEM and no longer the acronym CESM as you mistakenly use on page 5 line 196
The final part may further discuss the impact of targeted therapies and precision medicine on mammography screening (please see: 10.1186/s13244-020-00905-3). While targeted therapies and precision medicine have improved the treatment of breast cancer, early detection through screening remains crucial for improving outcomes and reducing mortality.
we have revised the paragraph on CEM according to your suggestions.
- Also, in my opinion it is really necessary to add a small paragraph on the role of the breast MRI to predict response to neoadjuvant chemotherapy and how can be used as a prognostic tool. Another area of research is the use of breast MRI for risk stratification in women with high genetic risk of breast cancer. As you know, breast MRI has great potential for predicting and prognosticating breast cancer (please see: 10.1186/s41747-022-00291-z).
As per your request, we have added a short paragraph on breast MRI. In particular, we have tried to focus on its possible role in screening high-risk patients.
- Conclusion
In my opinion, you can still elaborate a few more steps and clarify that with the advent of targeted therapies and precision medicine, the role of mammography in breast cancer screening and diagnosis may continue to evolve in the coming years. In any case, as you have amply described, mammography remains a critical tool in the early detection of breast cancer, and ongoing research efforts are focused on improving its performance and expanding its potential applications.
Thank you very much for the review.
We gladly re-read the articles you pointed out (many of which we already knew) and tried to revise the article according to all your suggestions.
We hope you will evaluate the article positively after our changes.
Reviewer 3 Report
The only saving grace to this manuscript is that it honestly calls itself an “Opinion”. There is so much misinformation in this manuscript that this reviewer wonders what personal experience they are basing their “opinion” on. To begin with, the title refers to “breast mammography”; is there any non-breast mammography? The authors present “overdiagnosis” both in the Abstract and in the Introduction as if it were an established fact, but appear to have based their opinion on a single publication by one author, without referring to any of the many, sophisticated analyses published in peer reviewed medical journals. This reviewer wonders why the authors have chosen that particular article to justify their own opinion. The promises in the Abstract remain promises because they are not delivered in the body of the manuscript. This reviewer would have expected the authors to focus their attention on the development and success of mammography screening as the first effective method for the early diagnosis of breast cancer. The authors failed to strike a balance between 50% reduction in breast cancer mortality achieved by organized mammography screening versus the statistical phenomenon of overdiagnosis, mounting to 1-5% in careful, long-term studies adjusting for lead time and the steadily increasing incidence of breast cancer. The single sentence citing reference 23 is not only untrue, but the reference is irrelevant: “Following the example of Sweden and the Netherlands, between 1986 and 1991, all major European countries started their own screening program (23)” This points on the part of the authors who failed to describe in detail the far more important subject of the randomized controlled mammography screening trials, which form the scientific basis for the introduction of population screening.
The following sentence uncovers the authors’ failure to understand the purpose of the randomized controlled trials (RCTs): “Sweden and Scotland were doing trials to confirm its use as a screening tool”. Instead, the RCTs were designed to investigate the impact of invitation to early detection with mammography upon mortality from breast cancer.
Besides, it is not only is the professional activity of Professor Charles Gros described in a demeaning way, but also his name is misspelled. Professor Dan Kopans founded "The Breast Imaging Division" in 1978 it what had been previously called the "Xeroradiography Division", long before their work on DBT. As for the MLO projection, it was developed by Dr. Bengt Lundgren and first described in 1974. The selection of references does not live up to the promise of the title.
Reviewer 4 Report
This is a short but very well written review of the history of breast imaging with focus on mammography. The text is excellent, the illustrations are of good quality. The reference list is correct and, where appropriate, it is up-to-date. I have two minor comments only.
- Line 30: replace the word “study” with the word “review” as it expresses the character of this manuscript more accurately
- The abbreviation FDA is explained in the line 218, although the abbreviation itself was also used earlier in the manuscript (line 185) without explanation.
The manuscript can be accepted for publication as it is, the proposed changes are more for copy editing.
Author Response
This is a short but very well written review of the history of breast imaging with focus on mammography. The text is excellent, the illustrations are of good quality. The reference list is correct and, where appropriate, it is up-to-date. I have two minor comments only.
- Line 30: replace the word “study” with the word “review” as it expresses the character of this manuscript more accurately
- The abbreviation FDA is explained in the line 218, although the abbreviation itself was also used earlier in the manuscript (line 185) without explanation.
The manuscript can be accepted for publication as it is, the proposed changes are more for copy editing.
We thank the reviewer's comments that gratify our work